# The lung microbiota in nontuberculous mycobacterial pulmonary disease

Bo-Guen Kim[1][⊚], Noeul Kang[1][⊚], Su-Young Kim[1], Dae Hun Kim[1]*, Hojoong Kim[1], O. Jung Kwon[1], Hee Jae Huh[2], Nam Yong Lee[2], Byung Woo Jhun[1]*

1 Division of Pulmonary and Critical Care Medicine, Department of Medicine, Samsung Medical Center, Sungkyunkwan University School of Medicine, Seoul, South Korea, 2 Department of Laboratory Medicine and Genetics, Samsung Medical Center, Sungkyunkwan University School of Medicine, Seoul, South Korea

⊚ These authors contributed equally to this work.
* byungwoo.jhun@gmail.com (BWJ); tnxo515@gmail.com (DHK)

**Data Availability Statement:** The raw data were registered in the Sequence Read Archive (SRA), and the SRA data (Bioproject.) They are available under the accession number PRJNA931790. The BioSamples are available under accession nos.

## Abstract

### Background

The role of bacterial microbiota in the pathogenesis of nontuberculous mycobacterial pulmonary disease (NTM-PD) is unclear. We aimed to compare the bacterial microbiome of disease-invaded lesions and non-invaded lung tissue from NTM-PD patients.

### Methods

We analyzed lung tissues from 23 NTM-PD patients who underwent surgical lung resection. Lung tissues were collected in pairs from each patient, with one sample from a disease-involved site and the other from a non-involved site. Lung tissue microbiome libraries were constructed using 16S rRNA gene sequences (V3–V4 regions).

### Results

Sixteen (70%) patients had *Mycobacterium avium* complex (MAC)-PD, and the remaining seven (30%) had *Mycobacterium abscessus*-PD. Compared to non-involved sites, involved sites showed greater species richness (ACE, Chao1, and Jackknife analyses, all p = 0.001); greater diversity on the Shannon index (p = 0.007); and genus-level differences (Jensen-Shannon, PERMANOVA p = 0.001). Analysis of taxonomic biomarkers using linear discriminant analysis (LDA) effect sizes (LEfSe) demonstrated that several genera, including *Limnohabitans*, *Rahnella*, *Lachnospira*, *Flavobacterium*, *Megamonas*, *Gaiella*, *Subdoligranulum*, *Rheinheimera*, *Dorea*, *Collinsella*, and *Phascolarctobacterium*, had significantly greater abundance in involved sites (LDA >3.00, p <0.05, and q <0.05). In contrast, *Acinetobacter* had significantly greater abundance at non-involved sites (LDA = 4.27, p<0.001, and q = 0.002). Several genera were differentially distributed between lung tissues from MAC-PD (n = 16) and *M. abscessus*-PD (n = 7), and between nodular bronchiectatic form (n = 12) and fibrocavitary form (n = 11) patients. However, there was no genus with a significant q-value.

SAMN33099469 (control) and SAMN33099470 (disease.).

**Funding:** This work was supported by the National Research Foundation of Korea (NRF) grant funded by the Korea government (MSIT) (2020R1C1C1008038). The funders were not involved in study design, data collection and analysis, decision to publish, or manuscript preparation.

**Competing interests:** The authors have declared that no competing interests exist.

## Conclusions

We identified differential microbial distributions between disease-invaded and normal lung tissues from NTM-PD patients, and microbial diversity was significantly higher in disease-invaded tissues.

## Trial registration

Clinical Trial registration number: NCT00970801.

## Introduction

Nontuberculous mycobacteria (NTM) are ubiquitous organisms found in natural environments and human communities. NTM infection is caused by complex interactions among immune status, microbiological factors, and environmental exposure. Most NTM infections lead to chronic pulmonary disease (PD), and the global burden of NTM-PD is increasing [1, 2]. Among more than 190 species of NTM, the *Mycobacterium avium* complex (MAC), mainly composed of *M. avium* and *M. intracellulare*, is the most common pathogen followed by *M. abscessus* in many countries [3].

The challenging aspect of NTM-PD management is that NTM-PD has a remarkably heterogeneous clinical course, unlike other respiratory infectious diseases [4, 5]. NTM-PD patients can show spontaneous negative conversion without antibiotic therapy or remain stable for several years. However, some patients exhibit rapid worsening of NTM-PD that involves cavity formation. Other obstacles include low treatment success rates, despite long-term multidrug antibiotic therapy [6]. In addition, clinicians commonly encounter mixed infections with various species of NTM, as well as repeated reinfection by different genotypes [7]. To date, there are limited data explaining the heterogeneity of the disease course or pathogenesis.

The microbiome, an ecological niche of microbial communities (e.g., bacteria, viruses, fungi, and archaea), has recently been reported to affect the clinical course and pathogenesis of respiratory diseases [8, 9]. Culture-independent techniques that involve targeted sequencing (e.g., analysis of the 16S ribosomal RNA [rRNA] gene) have been beneficial for detecting and identifying microbes. Recent studies of respiratory samples from patients with cystic fibrosis, bronchiectasis, or pulmonary fibrosis revealed differences in microbiota composition (i.e., dysbiosis), depending on the clinical course of the disease [10–12].

To explain the heterogeneity of the disease course or pathogenesis of NTM-PD, microbiome studies have been performed using various respiratory samples (e.g., alveolar fluid, oral washings, and sputum), and they have suggested the involvement of several dominant bacterial taxa [13–16]. However, these studies had some limitations, such as between-study discrepancies, possible sample contamination, insufficient sample sizes, or difficulties in the selection of controls. Therefore, the role of the microbiota in NTM-PD pathogenesis remains unclear. In this study, we conducted microbiome analysis using lung tissue specimens obtained under sterile conditions from NTM-PD patients who had undergone surgical lung resection. We compared the bacterial microbiome of disease-invaded lesions to non-invaded lung tissue in each patient. We also compared differences in bacterial distribution among patients with different NTM species or radiological phenotypes of NTM-PD. Our findings are useful for the characterization of microbial environments in the lung tissue of patients with NTM-PD.

## Methods

### Study participants

We screened consecutive NTM-PD patients who underwent adjuvant surgical lung resection, in addition to antibiotic therapy, between July 2012 and February 2019 at Samsung Medical Center, Seoul, South Korea (**S1 Fig**). We identified 40 patients with NTM-PD who consented to the collection of lung tissue. Lung tissue was collected in pairs from each patient, with one sample taken from a disease-invaded lesion and the other taken from normal tissue. We initially attempted to evaluate both bacterial and fungal microbiomes. A bacterial microbiome library was constructed using the V3-V4 region of 16S rRNA gene [17]. However, library construction failed with samples from 17 patients because of quality control (QC) failures or low valid read count (<200). Although a fungal microbiome library was constructed using the ITS 2 region [18], only one sample had a valid read count that allowed microbiome analysis. Therefore, in this study, fungal microbiome analysis was excluded.

Finally, the paired lung tissues of 23 NTM-PD patients were included in the analysis (**S1 Table**) and only the bacterial microbiomes were examined. The median duration of antibiotic therapy before surgery was 22.7 (interquartile range, 12.6–48.4) months. Details of antibiotics administered are shown in **S2 and S3 Tables**. We compared the microbiome data of patients according to disease involvement, NTM species, and radiological phenotype. Data were obtained from an ongoing prospective observational cohort (ClinicalTrials.gov identifier: NCT00970801). Written informed consent was obtained from the study participants and the Institutional Review Board of the Samsung Medical Center approved this study (IRB no. 2012–05–001).

### Lung tissue specimens

Lung tissue specimens were obtained under sterile conditions in the operating room from surgical material of patients who underwent lobectomy or pneumonectomy for NTM-PD. Specimens were acquired from two distinct sites in each patient: i) a cavity wall lesion in fibrocavitary (FC) form or a bronchiectasis lesion in nodular bronchiectatic (NB) form (hereafter, involved site), and ii) a site that appeared grossly normal without radiological involvement of NTM-PD from the resected lung (hereafter, non-involved site). These lung tissues were immediately dissected into multiple segments with a diameter of 5 mm. Dissected samples were snap-frozen in liquid nitrogen and stored at -80˚C.

### DNA extraction from lung tissue and MiSeq sequencing

DNA extraction from lung tissue was performed using a Maxwell Ⓡ RSC PureFood GMO and Authentication Kit (Promega, Madison, WI, USA) in accordance with the manufacturer's protocol. The concentration and quality of the extracted DNA were measured using an Epoch™ Spectrometer (BioTek, Winooski, VT, USA).

To analyze the bacterial microbiome of the lung tissue DNA, region V3 to V4 from the bacterial 16S rRNA gene was amplified using primers 341F (`TCGTCGGCAGCGTCAGATGTGTA TAAGAGACAGCCTACGGGNGGCWGCAG`) and 805R (`GTCTCGTGGGCTCGGAGATGTGTATA AGAGACAGGACTACHVGGGTATCTAATCC`). Amplification was performed in a C1000 Touch thermal cycler polymerase chain reaction (PCR) system (Bio-Rad, Hercules, CA, USA). The PCR cycling conditions were as follows: initial denaturation at 95˚C for 3 min; 25 cycles of denaturation at 95˚C for 30 s, annealing at 55˚C for 30 s, and elongation at 72˚C for 30 s; and final extension at 72˚C for 5 min.

Then, in accordance with the MiSeq system protocol for preparation of a 16S metagenomics sequencing library, a second PCR (Index PCR) was performed using a Nextera XT index kit (Illumina, San Diego, CA, USA) to attach an index code and Illumina sequencing adapter to the products of the first PCR. The second PCR cycling conditions were as follows: initial denaturation at 95˚C for 3 min; 8 cycles of denaturation at 95˚C for 30 s, annealing at 55˚C for 30 s, and elongation at 72˚C for 30 s; and final extension at 72˚C for 5 min.

The final PCR products were purified using a QIAquick PCR purification kit (Qiagen, Valencia, CA, USA) and quantified using a Quant-iT PicoGreen dsDNA Assay Kit (Promega). The quality of the final library was measured using a Bioanalyzer 2100 (Agilent, Palo Alto, CA, USA). Samples that did not pass QC during the DNA extraction and library production steps were excluded from further analysis. Libraries that passed QC were sequenced by CJ Bioscience, Inc. (Seoul, Korea) using a MiSeq Reagent Kit v2 (500-cycles) based on the Illumina MiSeq sequencing platform (Illumina) in accordance with the manufacturer's instructions. The raw data were registered in the Sequence Read Archive (accession number PRJNA931790).

## Sequence analysis

Microbiome profiling was conducted using the 16S-based Microbial Taxonomic Profiling (MTP) platform of the EzBioCloud application with the 16S database version PKSSU.4.0 [19]. All raw sequencing data were re-analyzed into each individual MTP using the EzBioCloud pipeline. MTP sets were constructed by grouping these individual MTPs. Comparative analysis between MTP sets was performed after normalization of gene copy numbers.

The relative abundance of sequences was compared between MTP sets using the Wilcoxon rank-sum test. Alpha (α) diversity was analyzed using the species richness (number of operational taxonomic units [OTUs], ACE, Chao1, and Jackknife) and the Shannon diversity index. Beta (β) diversity was evaluated using Jensen-Shannon divergence and generalized UniFrac distances, and the results were visualized by principal component analysis. Permutational multivariate analysis of variance (PERMANOVA) was used to analyze statistical differences in beta diversity. To identify differentially distributed taxa between MTP sets, which could potentially be used as microbial biomarkers, we performed taxonomic biomarker analysis based on linear discriminant analysis (LDA) effect sizes (LEfSe) and phylogenetic investigation of communities through the reconstruction of unobserved state algorithms [20, 21]. Taxa that demonstrated significant results in both relative abundance ($p < 0.05$) and LEfSe analysis (LDA score $>3.00$) were considered significant biomarkers. Statistical significance was set at p $<0.05$. Hotelling's t-test was used to compare bacterial profiles among categories. To correct for the false discovery rate, q-values were computed after correction for multiple tests to identify significant bacterial taxa in each category [22].

## Results

### Study participants

The characteristics of the 23 patients at the time of lung resection are shown in **Table 1**. The median age was 58 years, and 16 (70%) patients were women. Sixteen patients had MAC-PD (nine *M. avium* and seven *M. intracellulare*), and the remaining seven patients had *M. abscessus* subsp. *abscessus* (hereafter, *M. abscessus*)-PD. Approximately half (48%) of the patients had positive results in acid-fast bacilli smears and exhibited FC disease. Two (8%) patients achieved culture conversion status at the time of surgery.

**Table 1. Baseline characteristics of study patients at the time of lung resection (n = 23).**

| Characteristics | Value |
| --- | --- |
| Age, years | 58 (53–63) |
| Female | 16 (70) |
| Body mass index, kg/m$^2$ | 21.4 (19.5–22.7) |
| Never smoker | 18 (78) |
| Underlying disease | |
| Previous pulmonary tuberculosis | 12 (52) |
| Obstructive pulmonary disease or asthma | 1 (4) |
| Chronic pulmonary aspergillosis | 2 (8) |
| Etiology | |
| *M. avium* | 9 (40) |
| *M. intracellulare* | 7 (30) |
| *M. abscessus* | 7 (30) |
| Microbiological data | |
| Positive acid-fast bacilli smear | 11 (48) |
| Positive liquid medium culture | 16 (70) |
| Positive solid medium culture | 16 (70) |
| Radiological form | |
| Nodular bronchiectatic form | 12 (52) |
| Without cavity | 10/12 |
| With cavity | 2/12 |
| Fibrocavitary form | 11 (48) |
| Reason for lung resection | |
| Persistent culture positivity while on antibiotics* | 21 (92) |
| Rapid radiological progression | 7 (30) |
| Symptomatic worsening | 3 (13) |
| Time from diagnosis to surgery, months | 34.3 (17.0–51.2) |
| Duration of antibiotics before lung resection, months | 22.7 (12.6–48.4) |

Data are presented as n (%) or the median (interquartile range).

*Two patients had culture negative conversion at the time of lung resection.

## Comparison of microbiota proportions between involved and non-involved sites

We compared the relative proportions of microbial taxa between involved and non-involved sites at the phylum and genus taxonomic levels (**Fig 1**). The same four phyla were dominant at involved and non-involved sites, albeit at different proportions: involved sites, *Proteobacteria* (39.9%), *Firmicutes* (22.2%), *Actinobacteria* (13.3%), and *Bacteroidetes* (12.4%); non-involved sites, *Proteobacteria* (41.2%), *Actinobacteria* (17.7%), *Firmicutes* (16.1%), and *Bacteroidetes* (13.3%). In relative abundance analysis, the proportion of *Verrucomicrobia* was significantly higher at involved sites than at non-involved sites (p = 0.010) (**S2 Fig**).

At the genus level, *Mycobacterium* (5.6%), *Bacteroides* (5.1%), *Sphingomonas* (3.6%), and *Limnohabitans* (3.4%) were the dominant genera at involved sites. At non-involved sites, *Prevotella* (4.6%), *Acinetobacter* (4.5%), *Mycobacterium* (4.5%), and *Sphingobium* (4.0%) were dominant. In relative abundance analysis, the proportions of *Rahnella* (p = 0.002), *Aquabacterium* (p = 0.005), *Oscillibacter* (p = 0.005), *Limnohabitans* (p = 0.003), *Ruminococcus* (p = 0.006), *Blautia* (p = 0.026), and *Faecalibacterium* (p = 0.011) were significantly higher at

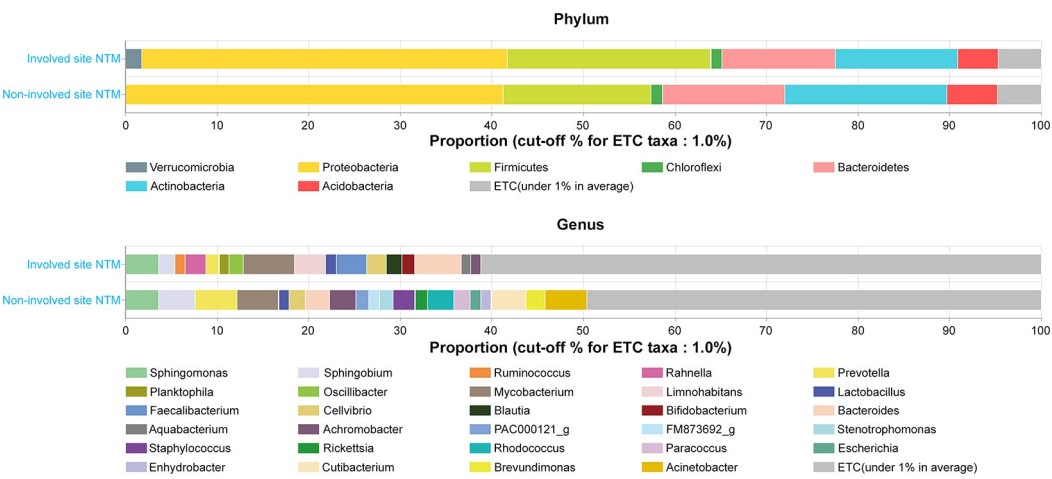

**Fig 1. Relative proportions of phyla and genera of lung microbiomes compared between involved and non-involved sites (taxa <1% are not indicated).**

involved sites than at non-involved sites, while the proportions of *Acinetobacter* (p<0.001) and *Enhydrobacter* (p = 0.022) were significantly higher at non-involved sites (**S3 Fig**).

## Major respiratory pathogens at the genus level

We compared the proportions of major respiratory pathogens at the genus level between involved and non-involved sites. Clinical microbiological data and proportions of *Mycobacterium* in each patient are shown in **Table 2**. The read counts tended to be higher at involved sites than at non-involved sites, but the differences were not statistically significant (p = 0.339). There were no correlations between clinical microbiological data (e.g., acid-fast bacilli smear and culture status results in sputum) and read counts of *Mycobacterium* in lung tissue. *Mycobacterium* was also detected at involved sites in two patients who achieved culture conversion in sputum samples at the time of lung resection (Mint-8 and Mint-10).

The read counts of other major respiratory pathogens (e.g., *Stenotrophomonas*, *Haemophilus*, *Burkholderia*, and *Pseudomonas*) are shown in **S4 Table**. *Burkholderia* was found in only one sample from a non-involved site, but there were no significant differences in the relative abundances of *Stenotrophomonas* (p = 0.583), *Haemophilus* (p = 0.733), or *Pseudomonas* (p = 0.423).

## Microbial diversities at involved and non-involved sites

The α-diversity of the microbiomes showed statistically significant differences between involved and non-involved sites. Involved sites had significantly higher numbers of OTUs when compared to non-involved sites (p = 2.9e-4), and showed greater richness in ACE, Chao 1, and Jackknife analyses (all p = 0.001). The Shannon index also indicated greater levels of diversity at involved sites (p = 0.007) (**Fig 2**).

In terms of β-diversity, there were clear genus-level differences between involved and non-involved sites (Jensen-Shannon PERMANOVA, p = 0.001) (**Fig 3**). Generalized UniFrac distances also indicated significant differences in genus-level differences between involved and non-involved sites (p = 0.001).

**Table 2. Proportions of *Mycobacterium* compared between involved and non-involved sites in study patients (n = 23).**

| Patient identifier | Clinical microbiological data from sputum | | | Mycobacterium in lung tissue | |
| --- | --- | --- | --- | --- | --- |
| | | | | Involved | Non-involved |
| | Acid-fast bacilli smear | Semi-quantitative culture* | Culture conversion | Read counts (%) | Read counts (%) |
| Mavi-1(FC) | – | trace | no | 55 (0.19) | – |
| Mavi-2(NB) | 2+ | trace | no | 5 (0.38) | 2 (0.24) |
| Mavi-3(FC) | – | liquid only | no | 190 (4.27) | – |
| Mavi-4(NB) | – | trace | no | 54 (0.73) | 2 (0.59) |
| Mavi-5(FC) | 2+ | 1+ | no | 6 (0.07) | 1 (0.09) |
| Mavi-7(FC) | – | trace | no | 21 (0.75) | – |
| Mavi-8(FC) | 1+ | trace | no | 2 (0.14) | – |
| Mavi-9(NB) | – | liquid only | no | – | 8 (0.68) |
| Mavi-10(NB) | trace | 1+ | no | – | – |
| Mint-2(NB) | trace | 1+ | no | 12 (0.16) | 3 (0.63) |
| Mint-4(FC) | 2+ | 1+ | no | 132 (1.22) | 2 (0.86) |
| Mint-5(FC) | – | liquid only | no | 86 (2.04) | – |
| Mint-6(FC) | – | 2+ | no | 23 (0.54) | 227 (63.06) |
| Mint-7(FC) | – | trace | no | 12 (0.13) | 1 (0.31) |
| Mint-8(NB) | – | – | yes | 13 (0.17) | – |
| Mint-10(FC) | – | – | yes | 969 (92.73) | – |
| Mabs-3(FC) | 3+ | 1+ | no | 3 (0.07) | 4 (0.72) |
| Mabs-4(NB) | trace | trace | no | 24 (0.21) | – |
| Mabs-5(NB) | – | trace | no | – | 4 (0.77) |
| Mabs-6(NB) | trace | liquid only | no | – | – |
| Mabs-7(NB) | trace | trace | no | – | – |
| Mabs-8(NB) | – | liquid only | no | – | – |
| Mabs-9(NB) | 4+ | 2+ | no | 6 (0.42) | 15 (1.63) |

FC, fibrocavitary; NB, nodular bronchiectatic;–, negative.

*negative, no growth in liquid or solid medium; liquid only, growth in liquid medium only; trace, growth of <50 colonies on solid medium; 1+, growth of 50 to 100 colonies on solid medium; 2+, growth of 100 to 200 colonies on solid medium; 3+, growth of 200 to 500 colonies on solid medium; and 4+, growth of >500 colonies on solid medium.

## Taxonomic biomarker analysis according to disease involvement

We performed LEfSe analysis to identify specific taxa that were differentially distributed between involved and non-involved sites. Taxa that showed significant results in both relative abundance and LEfSe analyses are shown in **Table 3**. The abundances of the phyla *Verrucomicrobia*, *Planctomycetes*, and *Nitrospirae* were higher at involved sites than at non-involved sites ($p < 0.05$). However, considering the cut-off value of the q-value, only the abundance of *Nitrospirae* was significantly higher at involved sites than at non-involved sites.

At the genus level, several genera, including *Limnohabitans*, *Faecalibacterium*, *Rahnella*, *Blautia*, *Ruminococcus*, *Oscillibacter*, *Lachnospira*, and *Methylotenera* were more abundant in involved sites than at non-involved sites. As many genera showed higher abundance at involved sites, only taxa ranking within the top 10 are described in Table 3 (the complete dataset is shown in **S5 Table**). Among all taxa found, the genera showing significant q-values were *Limnohabitans*, *Rahnella*, *Lachnospira*, *Flavobacterium*, *Megamonas*, *Gaiella*, *Subdoligranulum*, *Rheinheimera*, *Dorea*, *Collinsella*, and *Phascolarctobacterium* (**S5 Table**).

However, only *Acinetobacter* and *Enhydrobacter* were abundant at non-involved sites in the complete dataset. Among them, *Acinetobacter* was predominant at non-involved sites with

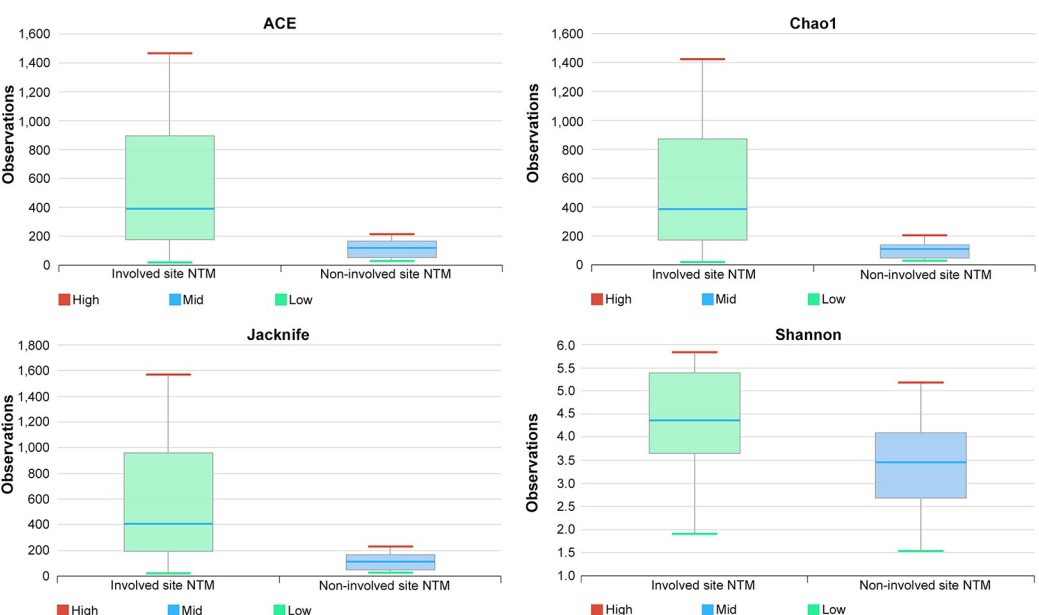

**Fig 2. Alpha-diversity compared between involved and non-involved sites.** Species richness (ACE p = 0.001, Chao1 p = 0.001, and Jackknife p = 0.001) and diversity index (Shannon p = 0.007), by Wilcoxon rank-sum test.

significant q-value. The detailed read counts of these genera in each patient are shown in **S6 Table**. Of them, only *Acinetobacter* showed a significant q-value.

## Taxonomic biomarker analysis according to NTM-PD etiology

Next, we identified specific genera that were differentially distributed between lung tissues from MAC-PD patients and *M. abscessus*-PD patients (**Table 4**). At involved sites, *Bacteroides*,

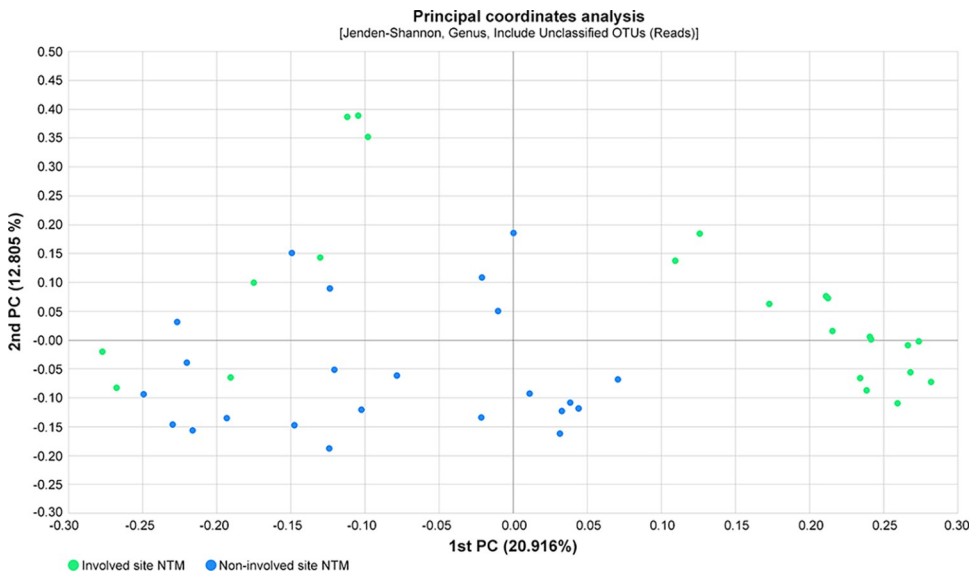

**Fig 3. Beta diversity distance compared between non-involved and involved sites in 23 NTM-PD patients (genus level, Jensen-Shannon analysis PERMANOVA p = 0.001).**

**Table 3. Taxonomic biomarker analysis for phyla and genera compared between involved and non-involved sites in study patients (n = 23).**

| Taxon name | Involved site (%) | Non-involved site (%) | LDA effect size | *p*-value | *q*-value[†] |
|---|---|---|---|---|---|
| **Phylum** | | | | | |
| *Verrucomicrobia* | 1.80 | 0.78 | 3.75 | 0.009 | 0.138 |
| *Planctomycetes* | 0.99 | 0.65 | 3.36 | 0.006 | 0.110 |
| *Nitrospirae* | 0.43 | 0.11 | 3.24 | < 0.001 | 0.027 |
| **Genus***  | | | | | |
| *Acinetobacter* | 0.64 | 4.53 | 4.27 | < 0.001 | 0.002 |
| *Limnohabitans* | 3.39 | 0.11 | 4.17 | < 0.001 | 0.027 |
| *Faecalibacterium* | 3.29 | 0.90 | 4.13 | 0.010 | 0.138 |
| *Rahnella* | 2.33 | 0.05 | 4.05 | < 0.001 | 0.029 |
| *Blautia* | 1.71 | 0.62 | 3.78 | 0.024 | 0.231 |
| *Ruminococcus* | 1.07 | 0.12 | 3.72 | 0.003 | 0.079 |
| *Oscillibacter* | 1.56 | 0.53 | 3.70 | 0.004 | 0.093 |
| *Lachnospira* | 0.98 | 0.11 | 3.67 | < 0.001 | 0.036 |
| *Methylotenera* | 0.97 | 0.07 | 3.60 | 0.003 | 0.085 |
| *Enhydrobacter* | 0.65 | 1.14 | 3.55 | 0.015 | 0.189 |

LEfSe, linear discriminant analysis effect size; LDA, linear discriminant analysis. LEfSe analysis included all taxa, including taxa with proportions <1%.

*Only the top 10 taxa are listed. Detailed data from the complete dataset are shown in online supplementary data.

[†]Adjusted p–value; the Benjamini–Hochberg false discovery rate was applied to correct for multiple testing, and values of less than 0.05 were considered significant.

*Rahnella*, *Streptococcus*, *Neisseria*, *Bacillus*, and *Agathobaculum* were significantly more abundant in tissues from MAC-PD patients. However, there was no genus with a significant q-value. There were no differences between MAC-PD and *M. abscessus*-PD patients in α-diversity at involved sites. However, there was a significant difference in β-diversity at these sites (Jensen-Shannon PERMANOVA, p = 0.013 and generalized UniFrac p = 0.023).

**Table 4. Taxonomic biomarker analysis for genera compared between MAC (n = 16) and *M. abscessus*-PD patients (n = 7).**

| Taxon name | MAC (%) | *M. abscessus* (%) | LDA effect size | *p*-value | *q*-value[†] |
|---|---|---|---|---|---|
| **Involved site** | | | | | |
| *Bacteroides* | 6.40 | 1.99 | 4.32 | 0.035 | 0.610 |
| *Rahnella* | 3.29 | 0.15 | 4.27 | 0.039 | 0.610 |
| *Methylotenera* | 0.05 | 3.07 | 4.13 | 0.022 | 0.610 |
| *Acidovorax* | 0.03 | 1.62 | 3.88 | 0.002 | 0.610 |
| *Rhizobacter* | 0.01 | 0.85 | 3.59 | 0.003 | 0.610 |
| *Undibacterium* | 0.03 | 0.70 | 3.50 | 0.031 | 0.610 |
| *Streptococcus* | 0.61 | 0.14 | 3.32 | 0.036 | 0.610 |
| *Neisseria* | 0.40 | 0.03 | 3.28 | 0.010 | 0.610 |
| *Bacillus* | 0.47 | 0.10 | 3.21 | 0.025 | 0.610 |
| *Rhodococcus* | 0.09 | 0.44 | 3.19 | 0.006 | 0.610 |
| *Agathobaculum* | 0.27 | 0.06 | 3.01 | 0.027 | 0.610 |
| **Non-involved site** | | | | | |
| *Staphylococcus* | 1.19 | 5.10 | 4.38 | 0.027 | 0.577 |
| *Bacteroides* | 1.82 | 4.49 | 4.03 | 0.016 | 0.577 |

PD, pulmonary disease; LEfSe, linear discriminant analysis effect size; LDA, linear discriminant analysis; MAC, *M. avium* complex. LEfSe analysis included all taxa, including taxa with proportions <1%. [†]Adjusted p–value; the Benjamini–Hochberg false discovery rate was applied to correct for multiple testing, and values of less than 0.05 were considered significant.

At non-involved sites, *Staphylococcus* and *Bacteroides* were more abundant in *M. abscessus*-PD patients than in MAC-PD patients. However, there were no differences in either α- or β-diversity between MAC-PD patients and *M. abscessus*-PD patients at non-involved sites. No genus had a significant q-value.

### Taxonomic biomarker analysis according to radiological phenotype

Finally, we evaluated differentially distributed genera according to the NTM-PD radiological phenotype (**Table 5**). At involved sites, several genera, including *Mycobacterium*, *Bacteroides*, *Faecalibacterium*, and *Blautia* were significantly more abundant in the FC form than in the NB form. As there were many genera showing a higher distribution in involved sites, only those ranked within the top five are described in **Table 5** (the complete dataset is shown in **S7 Table**). There was no genus with a significant q-value. There were no differences in α-diversity between these forms at involved sites. However, there was a significant difference in β-diversity between the two forms (Jensen-Shannon PERMANOVA p = 0.005 and generalized UniFrac p = 0.009).

At non-involved sites, *Escherichia* and *Streptococcus* were significantly more abundant in the FC form than in the NB form. There was no genus with a significant q-value. There were also no differences in either α- or β-diversity between the two forms.

## Discussion

We performed microbiome analysis using pairs of disease-invaded and non-invaded lung tissue from NTM-PD patients and identified several predominant taxa according to disease involvement. One of the most notable findings was that *Acinetobacter* and *Enhydrobacter* were more abundant at non-involved sites of NTM-PD lung tissues than the involved sites. In particular, *Acinetobacter* was predominant at non-involved sites even after correcting for false-positive rates. These findings imply that perturbations in microbial communities occur in the NTM-PD and also suggest that it is important to evaluate the relationships between the reduced presence of specific taxa, such as *Acinetobacter*, and deterioration of NTM-PD.

**Table 5. Taxonomic biomarker analysis for genera compared between the NB (n = 12) and FC (n = 11) forms.**

| Taxon name | NB form (%) | FC form (%) | LDA effect size | *p*-value | *q-v*alue[†] |
|---|---|---|---|---|---|
| **Involved site**[*] | | | | | |
| *Mycobacterium* | 0.42 | 11.24 | 4.58 | 0.018 | 0.492 |
| *Bacteroides* | 1.72 | 8.69 | 4.58 | 0.009 | 0.492 |
| *Faecalibacterium* | 1.10 | 5.69 | 4.39 | 0.008 | 0.492 |
| *Aquabacterium* | 1.94 | 0.03 | 3.99 | 0.023 | 0.492 |
| *Blautia* | 0.90 | 2.59 | 3.96 | 0.016 | 0.492 |
| **Non-involved site** | | | | | |
| *Bacteroides* | 4.00 | 1.13 | 4.10 | 0.009 | 0.429 |
| *Escherichia* | 0.19 | 2.27 | 4.04 | 0.022 | 0.429 |
| *Streptococcus* | 0.44 | 0.86 | 3.43 | 0.038 | 0.429 |
| *Clostridium* | 0.30 | 0.11 | 3.19 | 0.043 | 0.429 |

LEfSe, linear discriminant analysis effect size; LDA, linear discriminant analysis; NB, nodular bronchiectatic; FC, fibrocavitary. LEfSe analysis included all taxa, including taxa with proportions <1%.

[*]Only the top five taxa are listed. Detailed data from the complete dataset are shown in online supplementary data.

[†]Adjusted p–value; the Benjamini–Hochberg false discovery rate was applied to correct for multiple testing, and values of less than 0.05 were considered significant.

*Acinetobacter* is a genus of aerobic Gram-negative coccobacilli, which can be part of the normal flora of the skin and human respiratory secretions but which can also cause various human infections, including pneumonia or wound infections [23]. Previous studies in patients with obstructive respiratory disease, pneumonia, COVID-19, and tuberculosis have reported increased abundances of *Acinetobacter* [24–26], whereas interstitial pneumonia patients exhibited a reduced abundance of *Acinetobacter* in their alveolar fluid compared to healthy volunteers [26]. However, recent microbiome studies have not identified *Acinetobacter* species in bronchial washing fluid or sputum of NTM-PD patients [14, 15, 27]. In contrast to these studies, in our study, *Acinetobacter* was more abundant at non-involved sites of lung tissues, suggesting that the abundance of *Acinetobacter* decreased with invasion of the lung by various other bacteria, including NTM, possibly due to a competitive relationship with other bacteria. Only one South Korean study has reported that *Enhydrobacter* abundance is decreased in bronchial washing fluid from NTM-PD patients, but they did not find significant clinical implications [16]. Further studies are required to evaluate whether *Acinetobacter* plays a role in the pathogenesis of NTM-PD, whether it exhibits antagonism against NTM, and how it affects the host's innate immune system.

Notably, we identified large numbers of genera predominantly distributed at the involved sites, including *Limnohabitans*, *Rahnella*, *Lachnospira*, *Flavobacterium*, *Megamonas*, *Gaiella*, *Subdoligranulum*, *Rheinheimera*, *Dorea*, *Collinsella*, and *Phascolarctobacterium*. Philley et al. recently reported that *Lachnospira* was predominant in circulating extracellular vesicles from NTM patients [15]. Previous studies have also mentioned a relation between these genera and respiratory or other diseases. *Megamonas*, *Collinsella*, and *Phascolarctobacterium* were reported to be more abundant in the intestines of tuberculosis patients than in normal subjects [28–30], and *Lachnospira* and *Flavobacterium* were more distributed in the intestines of asthma and obese patients, respectively [31, 32]. A recent study showed that *Subdoligranulum* was more predominant in the lung tissue of patients with sub-solid-nodule lung cancer than with solid-nodule lung cancer [33]. However, there is still limited evidence regarding the associations of these genera with pathogenesis of NTM-PD. The abovementioned phenomena are assumed to be due to the influx of various bacteria from the outside due to the severe destruction of lung parenchyma during the course of NTM-PD. Therefore, it is possible that the genera predominantly found in the disease-invaded sites of NTM-PD may have little direct influence on NTM-PD pathogenesis, and some may be simply bystanders. Therefore, longitudinal assessment is necessary to evaluate significant perturbations in the microbial community of NTM-PD.

We identified different taxa distributions according to the causative NTM species and detected significant differences in β-diversity between MAC- and *M. abscessus*-invaded lung tissues. *Bacteroides*, *Rahnella*, *Streptococcus*, *Neisseria*, *Bacillus*, and *Agathobaculum* were abundant in MAC. There are no previous reports on comparative microbiome analysis according to NTM species. In previous studies using alveolar fluid or sputum, *Bacteroides* and *Bacillus* were predominant in the NTM group, and *Streptococcus* and *Neisseria* were observed in both NTM and control groups [14, 15, 27]. However, in our study, the differences in distributions of the above taxa were not significant after adjusting for the false-positive rate. Further studies are needed to identify more favorable bacterial environments for the survival of specific NTM species.

The clinical phenotypes of NTM-PD are divided into the FC form and NB form, the so-called "*Lady Windermere syndrome.*" The latter cases typically present with bronchiectasis in the middle lung zones and have unique body morphotypes, including lower body mass index and body fat percentage or taller stature [34]. Previous studies have focused on factors related to the typical NTM-PD phenotype in terms of immunological or metabolic factors [35, 36].

However, no studies have involved analysis of the microbiome according to NTM-PD phenotype. There were more dominant genera at involved sites in the FC form than the NB form (Table 5). In addition, proportions of *Mycobacterium* were higher in the FC form (11.24% vs. 0.42%, respectively). These findings suggest that the bacterial burden in the FC form is much higher than that in the NB form, and can be an evidence for the disease course of the FC form may be more aggressive than the NB form. In contrast, the predominant genera at non-involved sites in the NB form were *Bacteroides* and *Clostridium*. Although the roles that these bacteria play in the pathogenesis of NTM-PD have not been elucidated, our data suggest that the microbial environment may vary according to phenotype. However, there were no taxa with a significant q-value, and several factors, such as low biomass in the lung tissue [37, 38] or limitations of the 16S rRNA technique [39], should be taken into consideration.

In our study, α-diversity was higher at involved sites than at non-involved sites. In contrast, disease exacerbation and decreased diversity have been reported in previous studies of other respiratory diseases, such as chronic obstructive pulmonary disease [40]. But, a recent microbiome study analyzed sputum from 42 NTM-PD patients and reported that the Shannon index and observed richness were higher in those receiving treatment for NTM [41]. These observations suggest that diversity increases as the disease worsens in NTM-PD and that worsening of NTM-PD is accompanied by an increase in the invasion of other bacteria. Further research of this issue is required.

Our study had several limitations. First, a large number of patients were excluded due to QC failures or low valid read counts, a possible consequence of the relatively low biomass in lung tissue compared to other organs. Similar phenomena have also been observed in previous studies that have used lung tissues from patients with other chronic respiratory diseases [37, 38]. Second, there are limitations to using the 16S rRNA gene sequencing to identify mycobacteria. Previous studies have reported a lack of mycobacterial identification by 16S rRNA gene sequencing in samples with positive cultures for these organisms [27, 39, 42]. Sequencing of this gene cannot distinguish between *M. tuberculosis* and NTM, and detailed species or subspecies identification of NTM cannot be performed. In addition, there may be taxa that have not been identified by 16S rRNA gene sequencing. Third, as we analyzed patients who had been treated with antibiotics for a long time, it is possible that the distribution of the microbiome had already been affected by the use of antibiotics. Fourth, this study was conducted only on Asian patients treated at a single institution. Therefore, further longitudinal analyses in larger numbers of patients are required to confirm our results.

In conclusion, we identified distinct microbial distributions between disease-invaded and non-invaded lung tissues from NTM-PD patients. Diversity was significantly higher in disease-invaded tissues.

## Supporting information

**S1 Fig. Study samples.**
(DOCX)

**S2 Fig. Taxonomic relative abundance of phyla compared between involved and non-involved sites.** Taxa <1% are not indicated.
(DOCX)

**S3 Fig. Taxonomic relative abundance of genera compared between involved and non-involved sites.** Taxa <1% are not indicated.
(DOCX)

**S1 Table. Library data on lung tissue samples included in the study.**
(DOCX)

**S2 Table. Antibiotics administered before lung resection of all patients (n = 40).**
(DOCX)

**S3 Table. Detailed information of antibiotics administered before lung resection of study patients (n = 23).**
(DOCX)

**S4 Table. Proportions of other major respiratory pathogens compared between involved and non-involved sites in study patients (n = 23).**
(DOCX)

**S5 Table. Taxonomic biomarker analysis for genera compared between involved and non-involved sites in study patients (n = 23).**
(DOCX)

**S6 Table. Predominant genera in non-involved site compared to involved site identified by LEfSe analysis (n = 23).**
(DOCX)

**S7 Table. Taxonomic biomarker analysis for genera compared between NB (n = 12) and FC (n = 11) forms in involved site.**
(DOCX)

## Acknowledgments

We would like to express our heartfelt gratitude and respect to Dr. Won-Jung Koh, for giving us invaluable guidance and unfailing support for NTM-PD research. Dr. Won-Jung Koh passed away in August 2019. We dedicate this work to his memory.

## Author Contributions

**Conceptualization:** Bo-Guen Kim, Noeul Kang, Su-Young Kim, Dae Hun Kim, O. Jung Kwon, Hee Jae Huh, Byung Woo Jhun.

**Data curation:** Bo-Guen Kim, Noeul Kang, Su-Young Kim, Dae Hun Kim, Byung Woo Jhun.

**Formal analysis:** Bo-Guen Kim, Byung Woo Jhun.

**Funding acquisition:** Byung Woo Jhun.

**Resources:** Hojoong Kim, O. Jung Kwon, Hee Jae Huh, Nam Yong Lee.

**Writing – original draft:** Bo-Guen Kim, Noeul Kang, Su-Young Kim, Dae Hun Kim, Byung Woo Jhun.

**Writing – review & editing:** Dae Hun Kim, Hojoong Kim, O. Jung Kwon, Hee Jae Huh, Nam Yong Lee, Byung Woo Jhun.

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
