## [Decision Letter · Decision Letter 0]

20 Feb 2023

PONE-D-23-01037The lung microbiota in nontuberculous mycobacterial pulmonary diseasePLOS ONE

Dear Dr. Jhun,

Thank you for submitting your manuscript to PLOS ONE. After careful consideration, we feel that it has merit but does not fully meet PLOS ONE’s publication criteria as it currently stands. Therefore, we invite you to submit a revised version of the manuscript that addresses the points raised during the review process. As commented by  the reviewers, the treatment information of patients and its impact on the interpretation of the study results should be mentioned and discussed. Also, the authors should take this opportunity to edit/revise this manuscript for any other errors, and highlight the significance of their work compared to similar work that were published in the past.

We look forward to receiving your revised manuscript.

Kind regards,

Selvakumar Subbian, Ph.D.

Academic Editor

PLOS ONE

Journal Requirements:

"This work was supported by the National Research Foundation of Korea (NRF) grant funded by the Korea government (MSIT) (2020R1C1C1008038)."

"No authors have competing interests."

Reviewers' comments:

Reviewer's Responses to Questions

**Comments to the Author**

1. Is the manuscript technically sound, and do the data support the conclusions?

Reviewer #1: Partly

Reviewer #2: Yes

2. Has the statistical analysis been performed appropriately and rigorously? 

Reviewer #1: I Don't Know

Reviewer #2: Yes

3. Have the authors made all data underlying the findings in their manuscript fully available?

Reviewer #1: No

Reviewer #2: No

4. Is the manuscript presented in an intelligible fashion and written in standard English?

Reviewer #1: Yes

Reviewer #2: Yes

5. Review Comments to the Author

Reviewer #1: Overall, this is an interesting study to gain insight in the lung microbiota in patients with NTM using direct lung tissue specimens. Some comments to consider:

Abstract

- Line 42-45 – if the q value is not significant for this finding, should consider if it’s overstating the genera were truly differentially distributed – this is something to note throughout the paper

Intro

- Line 112-113 – this study provides characterization of the lung microbiota but does not delve into determining the pathogenesis of NTM pulmonary disease from a microbiome view. I believe studies such as this is an important key step but would consider reordering this statement so it’s more reflective of the inferences the results truly provide.

Methods

- Line 124 – 17 of 40 patients having QC failure or low read counts is notable and a substantial proportion. Any further details particularly on the QC failures could be useful as it doesn’t seem by random chance.

- Last paragraph before Results section – should add in statistical approach to adjusting for multiple hypothesis testing and use of q value.

Results

- Line 256 – based on table 3 and the q- value, two phyla - Verrucomicrobia, Planctomycetes - and not significantly different between involved and non-involved sites. The table appropriately shows the q value as the B-H adjusted p value but the text in this line is not consistent with that conclusion. Also, the use of adjusted value using the B-H method should be in the methods section (as noted above).

- Line 266-268 – same issue as above – it is noted Acinetobacter and Enhydrobacter is significantly more abundant than in non-involved sites but then said only Acinetobacter based on the q value – the significance of using p value for the former statement is misleading given the issues of false discoveries (which is why q value is used). And same framing issue of significance with p value vs q value in the next two section – analysis by NTM etiology and radiological phenotype.

Discussion

- First paragraph – the results are being overstated here as discussed above. Enhydrobacter is not significantly different in abundance based on the appropriate significance measure – adjusted p value or q value. It is framed as if adjusting the p value for false discoveries is optional vs what needs to be done with multiple hypothesis testing. Additionally, this is a cross sectional study with no longitudinal follow up or initial microbiome analysis early in NTM disease. So it is unknown what is happening to the microbiota in the “process of NTM-PD”.

- Second paragraph on Acinetobacter – any discussion on how the antibiotics these patients were impacted the lung microbiota profile in non involved areas? Do you think (or any literature) the Acinetobacter abundance would differ in a patient naïve to or off antibiotics for a while?

- Line 325-328 – were all these genera significant after adjusting the p value?

- Any discussion on the characteristics of the sample studied? Interesting that a large portion of the sample had previous pulmonary TB and that was their underlying condition.

Reviewer #2: Kim and collaborators have analyzed the microbiome of disease-invaded lesions and non-invaded lung tissue from nontuberculous mycobacterial pulmonary disease patients finding that microbial diversity was significantly higher in disease-invaded tissues.

The manuscript is well written and the methodology is clear. Some concerns and suggestions follow:

Authors state that a drawback from their study is that they analyzed patients who had been treated with antibiotics for a long time. Information on treatments is in the supplementary result section. I beleive that the fact they recieved antimicrobial treatment previous to lung resection should be mentioned in the M&M section and not in the results sections lines 204-205. The table is informative, however, it is expressed as percentages. I think it would be helpful to have the raw data on what treatment each patient recieved, even the patients excluded from the study due to QC failure for library preparation or exclusion due to low value read. Did authors look into the effect of this treatments? Is there a correlation between these treatments and the microbial findings?

I am surprised by the results of the study. I would have expected to find less diversity in the NTM involved site. In Lines 369-371, authors do cite a study in this line, this is, disease exacerbation and decreased diversity related to other respiratory infections. Also, since Acinetobacter is nosocomial and opportunistic I would have expeceted to find it in both types of samples and not exclusively in the non NTM involved site. Do authors have a theory on why they have obtained these results?

Line 382-383 A reference is needed after the following statement “Previous studies have reported a lack of mycobacterial identification by 16S rRNA gene sequencing in samples with positive cultures for these organisms.”

6. PLOS authors have the option to publish the peer review history of their article (what does this mean?). If published, this will include your full peer review and any attached files.

Reviewer #1: No

Reviewer #2: No

---

## [Author Response · Author response to Decision Letter 0]

23 Mar 2023

Journal Requirements

Comment (C) 1. Please ensure that your manuscript meets PLOS ONE's style requirements, including those for file naming.

Response (R) 1. Thank you for the suggestion. We have revised the manuscript accordingly.

C2. Please provide additional details regarding participant consent. In the ethics statement in the Methods and online submission information, please ensure that you have specified what type you obtained (for instance, written or verbal, and if verbal, how it was documented and witnessed). If your study included minors, state whether you obtained consent from parents or guardians. If the need for consent was waived by the ethics committee, please include this information.

R2. Written informed consent was obtained from the study participants. We have added the details of participant consent in the Methods section (lines 112–113).

C3. We note that the grant information you provided in the ‘Funding Information’ and ‘Financial Disclosure’ sections do not match. When you resubmit, please ensure that you provide the correct grant numbers for the awards you received for your study in the ‘Funding Information’ section.

R3. We have added this information to the revised manuscript.

C4. Thank you for stating the following financial disclosure: "This work was supported by the National Research Foundation of Korea (NRF) grant funded by the Korea government (MSIT) (2020R1C1C1008038)." Please state what role the funders took in the study. If the funders had no role, please state: "The funders had no role in study design, data collection and analysis, decision to publish, or preparation of the manuscript." If this statement is not correct you must amend it as needed. Please include this amended Role of Funder statement in your cover letter; we will change the online submission form on your behalf.

R4. Thank you for pointing this out. The funders were not involved in study design, data collection and analysis, decision to publish, or manuscript preparation. We have added this information to the revised manuscript (lines 414–415).

C5. Thank you for stating the following in your Competing Interests section: "No authors have competing interests."

R5. We have revised this information in the revised manuscript (line 418).

C6. In your Data Availability statement, you have not specified where the minimal data set underlying the results described in your manuscript can be found. PLOS defines a study's minimal data set as the underlying data used to reach the conclusions drawn in the manuscript and any additional data required to replicate the reported study findings in their entirety. All PLOS journals require that the minimal data set be made fully available.

R6. The raw data were registered in the Sequence Read Archive (SRA), and the SRA data (Bioproject) are available under the accession number PRJNA931790. We have added this information to the revised manuscript (lines 154–155). The BioSamples will be released under accession nos. SAMN33099469 (control) and SAMN33099470 (disease) on May 1, 2023.

C7. Please include captions for your Supporting Information files at the end of your manuscript, and update any in-text citations to match accordingly.

R7. We have revised it.

C8. Please review your reference list to ensure that it is complete and correct. If you have cited papers that have been retracted, please include the rationale for doing so in the manuscript text, or remove these references and replace them with relevant current references. Any changes to the reference list should be mentioned in the rebuttal letter that accompanies your revised manuscript. If you need to cite a retracted article, indicate the article’s retracted status in the References list and also include a citation and full reference for the retraction notice.

R8. We have reviewed the reference list and made the necessary changes. 

Response to reviewer

Reviewer #1

Overall, this is an interesting study to gain insight in the lung microbiota in patients with NTM using direct lung tissue specimens. Some comments to consider:

Abstract

C1. Line 42-45 – if the q value is not significant for this finding, should consider if it’s overstating the genera were truly differentially distributed – this is something to note throughout the paper.

R1. Thank you for your comments. It is essential to consider the possibility of false positive results in the microbiome analysis. Thus, we computed q-values to evaluate the accuracy of our results, which have been added to the text and tables. The q-values enable more rigorous microbiome analysis.

Intro

C2. Line 112-113 – this study provides characterization of the lung microbiota but does not delve into determining the pathogenesis of NTM pulmonary disease from a microbiome view. I believe studies such as this is an important key step but would consider reordering this statement so it’s more reflective of the inferences the results truly provide.

R2. Thank you for your comment. We have revised the sentence in the Introduction to the following: “Our findings are useful for the characterization of microbial environments in the lung tissue of patients with NTM-PD.” (lines 90–91).

Methods

C3. Line 124 – 17 of 40 patients having QC failure or low read counts is notable and a substantial proportion. Any further details particularly on the QC failures could be useful as it doesn’t seem by random chance.

R3. Thank you for your comment. As shown in S1 Fig, certain tissue samples had read values with low validity (< 200) and were thus excluded from the analysis. Values with low validity have also been observed in other microbiome studies of lung tissue (1,2), probably due to the small lung biomass. This phenomenon occurs due to destruction of the lung and its parenchyma by chronic lung disease.

We compared clinical characteristics between the ‘study population’ (n = 23) and the ‘excluded population’ due to QC failure or low read counts (n = 17). The median duration of antibiotic therapy before surgery was 22.7 (interquartile range, 12.6–48.4) months in the study population and was 16.8 (interquartile range, 12.3–29.4) months in the excluded population (p = 0.374). The etiologies of NTM-PD in the excluded population were M. avium (n = 3, 18%), M. intracellulare (n = 4, 24%), and M. abscessus (n = 10, 58%), with no significant difference between the excluded and study populations (p = 0.171). The two groups had no significant differences in terms of the antibiotic type or duration. We have added this information to S2 Table. In the present study, failure to extract high-quality data was not because of prolonged antibiotic use.

(1) Microbiome in lung explants of idiopathic pulmonary fibrosis: a case-control study in patients with end-stage fibrosis, Thorax. 2018;73:481–4.

(2) Pulmonary bacterial communities in surgically resected noncystic fibrosis bronchiectasis lungs are similar to those in cystic fibrosis, Pulm Med. 2012;2012:746358.

C4. Last paragraph before Results section – should add in statistical approach to adjusting for multiple hypothesis testing and use of q value.

R4. Thank you for your suggestion. We have added the details of the statistical analysis and q-value in the Methods section, as follows: “Hotelling's t-test was used to compare bacterial profiles among categories. To correct for the false discovery rate, q-values were computed after correction for multiple tests to identify significant bacterial taxa in each category” (lines 176–179).

Results

C5. Line 256 – based on table 3 and the q- value, two phyla - Verrucomicrobia, Planctomycetes - and not significantly different between involved and non-involved sites. The table appropriately shows the q value as the B-H adjusted p value but the text in this line is not consistent with that conclusion. Also, the use of adjusted value using the B-H method should be in the methods section (as noted above).

R5. We described based on the p-value in the original manuscript. This was because we wanted to explain the tendency shown in our results without strictly satisfying the cut-off of the q-value. As you pointed out, we have revised it, as follows ‘However, considering the cut-off value of the q-value, only the abundance of Nitrospirae was significantly higher at involved sites than at non-involved sites’ (line 241-242).

C6. Line 266-268 – same issue as above – it is noted Acinetobacter and Enhydrobacter is significantly more abundant than in non-involved sites but then said only Acinetobacter based on the q value – the significance of using p value for the former statement is misleading given the issues of false discoveries (which is why q value is used). And same framing issue of significance with p value vs q value in the next two section – analysis by NTM etiology and radiological phenotype.

R6. Thank you for your comment. We have revised it, as follows ‘However, only Acinetobacter and Enhydrobacter were abundant at non-involved sites in the complete dataset. Among them, Acinetobacter was predominant at non-involved sites with significant q-value’ (line 251-253).

Discussion

C7. First paragraph – the results are being overstated here as discussed above. Enhydrobacter is not significantly different in abundance based on the appropriate significance measure – adjusted p value or q value. It is framed as if adjusting the p value for false discoveries is optional vs what needs to be done with multiple hypothesis testing. Additionally, this is a cross sectional study with no longitudinal follow up or initial microbiome analysis early in NTM disease. So it is unknown what is happening to the microbiota in the “process of NTM-PD”.

R7. We wanted to highlight the changing pattern of bacterial distribution. Additionally, we discussed the possibility of false-positive results. So we have already mentioned that ‘In particular, Acinetobacter was predominant at non-involved sites even after correcting for false-positive rates." Another reason for mentioning Enhydrobacter is that Enhydrobacter was mentioned by another research team in the past. This is briefly described in the discussion ‘Only one South Korean study has reported that Enhydrobacter abundance is decreased in bronchial washing fluid from NTM-PD patients, but they did not find significant clinical implications’. Additionally, we have removed the term “process” to clarify our meaning (line 291).

C8. Second paragraph on Acinetobacter – any discussion on how the antibiotics these patients were impacted the lung microbiota profile in non-involved areas? Do you think (or any literature) the Acinetobacter abundance would differ in a patient naïve to or off antibiotics for a while?

R8. Thank you for your insightful comment. Our study was a cross-sectional design, involving participants who needed surgical resection despite long-term antibiotics. We hypothesized that there is a difference in the distribution of Acinetobacter before and after antibiotic use, which should be evaluated in a longitudinal study. As described in Discussion section, changes in the distribution of "Acinetobacter" have been confirmed in other lung diseases, however, the present study is the first to report these findings in NTM-PD. The difference in Acinetobacter distribution before and after antibiotic use may be due to bacterial antagonism, antibiotic pressure, or immune reaction with more virulent bacterial species. However, these hypotheses have not been discussed in the Discussion due to a lack of evidence. Further studies are needed.

C9. Line 325-328 – were all these genera significant after adjusting the p value?

R9. Please see R1 and S5 Table.

C10. Any discussion on the characteristics of the sample studied? Interesting that a large portion of the sample had previous pulmonary TB and that was their underlying condition.

R10. South Korea has an intermediate incidence of tuberculosis, which was shown in our previous studies (1–3). Additionally, sequelae of tuberculosis predispose to the development of NTM-PD. As a result, some NTM-PD patients in South Korea have a history of pulmonary tuberculosis. In particular, these patients have sequelae of tuberculosis due to fibrotic changes after treatment, which predispose them to the development of FC or refractory NTM-PD.

(1) Mortality and Prognostic Factors of Nontuberculous Mycobacterial Infection in Korea: A Population-based Comparative Study, Clin Infect Dis 2021;72(10):e610-e619

(2) Impact of Time Between Diagnosis and Treatment for Nontuberculous Mycobacterial Pulmonary Disease on Culture Conversion and All-Cause Mortality, Chest 2022;161(5):1192-1200

(3) Prognostic factors associated with long-term mortality in 1445 patients with nontuberculous mycobacterial pulmonary disease: a 15-year follow-up study, Eur Respir J 2;55(1):1900798 

Reviewer #2

Kim and collaborators have analyzed the microbiome of disease-invaded lesions and non-invaded lung tissue from nontuberculous mycobacterial pulmonary disease patients finding that microbial diversity was significantly higher in disease-invaded tissues. The manuscript is well written and the methodology is clear. Some concerns and suggestions follow:

C1. Authors state that a drawback from their study is that they analyzed patients who had been treated with antibiotics for a long time. Information on treatments is in the supplementary result section. I believe that the fact they received antimicrobial treatment previous to lung resection should be mentioned in the M&M section and not in the results sections lines 204-205.

R1. Thank you for your suggestion. We have moved the information about the use of antibiotics from the Results section to the Methods section (lines 107-109).

C2. The table is informative; however, it is expressed as percentages. I think it would be helpful to have the raw data on what treatment each patient received, even the patients excluded from the study due to QC failure for library preparation or exclusion due to low value read. Did authors look into the effect of this treatments? Is there a correlation between these treatments and the microbial findings?

R2. Thank you for your comment. We described raw data for all analyses, except taxonomic biomarker analysis (LEfSe analysis). This is because, in the taxonomic biomarker analysis, “normalization” was performed to correct the read count variation in each sample for the effect size. We believe that the addition of raw data for the taxonomic biomarker analysis would lead to confusion.

 In addition, we presented the raw data of detailed antibiotic treatment history in supplementary data (S3 Table). Also, we compared the difference in treatment history between patients who were excluded from this study due to inadequate quality data and patients who belonged to this study (S2 Table). There was no significant difference in the type or duration of antibiotics used between the two groups. We added this information in S2 Table. The failure to extract data of sufficient quality for analysis through this is not because of the long use of antibiotics. We estimate that a significant number of samples showed QC failure or low read count due to the low biomass of lung tissue itself. These factors should be considered in future studies of microbiome analysis of lung tissue.

C3. I am surprised by the results of the study. I would have expected to find less diversity in the NTM involved site. In Lines 369-371, authors do cite a study in this line, this is, disease exacerbation and decreased diversity related to other respiratory infections. Also, since Acinetobacter is nosocomial and opportunistic I would have expected to find it in both types of samples and not exclusively in the non NTM involved site. Do authors have a theory on why they have obtained these results?

R3. Thank you for your comment. We expected similar results as you. However, there was higher diversity in the involved sites than in the uninvolved sites, which may have been because diversity increases with increasing NTM-PD severity and because NTM-PD worsening is accompanied by increased invasion of other bacteria (as described in the Discussion). These changes were observed because we used lung tissue, rather than sputum. Because lung has a lower biomass than other organs, diversity increases temporarily when NTM invades and destroys the lung parenchyma. However, similar changes may not be observed if sputum is analyzed. Further longitudinal studies are needed on this topic. We have added an additional reference (reference no. 40) to support this discussion, although it is related to other respiratory diseases.

The reasons underlying the difference in the Acinetobacter distribution are unclear. However, Acinetobacter can be a component of the normal flora of human skin and respiratory secretions, as well as a pathogen in human infections, including pneumonia and wound infections. Several hypotheses may be proposed to explain our findings. First, destruction of the lung parenchyma by NTM may make the environment unviable for Acinetobacter. Second, NTM and Acinetobacter may have an antagonistic relationship; therefore, Acinetobacter may be distributed preferentially in NTM-uninvolved sites. Third, Acinetobacter can stimulate T cells to promote the eradication of NTM by alveolar macrophages. Further studies are needed to provide evidence for these hypotheses.

C4. Line 382-383 A reference is needed after the following statement “Previous studies have reported a lack of mycobacterial identification by 16S rRNA gene sequencing in samples with positive cultures for these organisms.”

R4. Thank you for your comment. We have added the appropriate reference.

---

## [Decision Letter · Decision Letter 1]

17 Apr 2023

The lung microbiota in nontuberculous mycobacterial pulmonary disease

PONE-D-23-01037R1

Dear Dr. Jhun,

We’re pleased to inform you that your manuscript has been judged scientifically suitable for publication and will be formally accepted for publication once it meets all outstanding technical requirements.

Kind regards,

Selvakumar Subbian, Ph.D.

Academic Editor

PLOS ONE

Additional Editor Comments (optional):

Reviewers' comments:

Reviewer's Responses to Questions

**Comments to the Author**

1. If the authors have adequately addressed your comments raised in a previous round of review and you feel that this manuscript is now acceptable for publication, you may indicate that here to bypass the “Comments to the Author” section, enter your conflict of interest statement in the “Confidential to Editor” section, and submit your "Accept" recommendation.

Reviewer #1: All comments have been addressed

2. Is the manuscript technically sound, and do the data support the conclusions?

Reviewer #1: Yes

3. Has the statistical analysis been performed appropriately and rigorously? 

Reviewer #1: I Don't Know

4. Have the authors made all data underlying the findings in their manuscript fully available?

Reviewer #1: Yes

5. Is the manuscript presented in an intelligible fashion and written in standard English?

Reviewer #1: Yes

6. Review Comments to the Author

Reviewer #1: (No Response)

7. PLOS authors have the option to publish the peer review history of their article (what does this mean?). If published, this will include your full peer review and any attached files.

Reviewer #1: No

---

## [Editor Report · Acceptance letter]

17 May 2023

PONE-D-23-01037R1 

The lung microbiota in nontuberculous mycobacterial pulmonary disease 

Dear Dr. Jhun:

I'm pleased to inform you that your manuscript has been deemed suitable for publication in PLOS ONE. Congratulations! Your manuscript is now with our production department. 

Kind regards, 

on behalf of

Dr. Selvakumar Subbian 

Academic Editor

PLOS ONE